# Ultrahigh-Pressure Mineral Inclusions in a Crustal Granite: Evidence for a Novel Transcrustal Transport Mechanism

**Rainer Thomas** [1,*], **Paul Davidson** [2], **Adolf Rericha** [3] **and Ulrich Recknagel** [4]

1   Im Waldwinkel 8, D-14662 Friesack, Germany
2   Codes, Centre for Ore Deposits and Earth Sciences, University of Tasmania, Hobart 7001, Australia
3   Alemannenstraße 4a, D-144612 Falkensee, Germany
4   Böhmerwaldstraße 22, D-86529 Schrobenhausen, Germany
*   Correspondence: rainerthomas@t-online.de

**Abstract:** Spherical crystals in minerals from prismatine-bearing rock from Waldheim, including ultrahigh-pressure (UHP) minerals such as stishovite and coesite, were previously described in uncommon crustal environments. To determine if this was an outlier phenomenon, we searched for equivalent inclusions in other rocks, which we indeed discovered in a Variscan tin-bearing granite sensu stricto from the Erzgebirge/Germany. The identification of more examples of this phenomenon implies a novel, very rapid transcrustal transport mechanism, which, however, is not unique. We demonstrate the unusual occurrence of UHP minerals (moissanite, diamond, lonsdaleite, stishovite, coesite, kumdykolite, and cristobalite-II) in topaz the investigated granitic samples, which reflects the direct interaction of mantle and crust via supercritical fluids or extremely volatile-rich melts. Mostly, the UHP minerals we recognized occur as tiny inclusions in moissanite. The trapping by this mineral prevents a fast reaction in an exogenous environment.

**Keywords:** UHP minerals; stishovit; coesite; diamond; lonsdaleite; supercritical fluid; granite; transcrustal transport





## 1. Introduction

Recently, the present authors intensively studied a prismatine-bearing rock from the Waldheim granulite using micro-Raman spectroscopy (see Appendix A) after careful microscopic investigation [1–4]. Significant findings of our studies include the discovery of water-rich stishovite inclusions in almandine and extremely water-rich coesite inclusions in prismatine [3,4]. In addition to other minerals, the occurrence of water-rich stishovite and coesite is uncommon in the pressure–temperature (P–T) regime of their host rocks, implying transport from extreme depths and incorporation into minerals in felsic host rocks such as granites in the crust. This is especially significant because the co-trapping as mineral inclusions seems to have prevented the inversion to lower-pressure forms, which is typical for most other transport mechanisms.

We observed tiny (size range 10 to 50 μm), very smooth, mostly monomineralic crystals of zircon, diamond, and moissanite with spherical habits, mainly in prismatine but also in other minerals, including corundum, tourmaline, garnet, sillimanite, and zircon. Kalkowsky [5] was the first to describe such smooth zircon spheres similar to oil drops but did not provide an interpretation for this observation. In addition, we observed complex spherical crystals containing other spheres, such as anorthite, in a spherical corundum matrix. It is suggested that the spherical crystals are the result of mechanical abrasion and possibly chemical corrosion during the transport in a fast-flowing supercritical fluid [3]. Sphere-in-sphere implies a multi-stage process. Furthermore, these smooth spherical crystals in the host mineral indicate that the host mineral's crystallization was so rapid that a polyhedric equilibrium crystal form could not be attained. These spherical crystals represent "foreign bodies" in the Waldheim prismatine. They show non-equilibrium

features, such as the absence of equilibrium faces and the lack of previously described post-entrapment shape modifications [6]. Water-rich stishovite and coesite are silica polymorphs, which crystallize under very high P–T conditions (>30 GPa and ~1000 °C [7]. In contrast, the P–T conditions of the prismatine horizon were determined to be ~1.3 GPa and 1000 °C [8,9], well below the stability fields of stishovite, coesite, and diamond.

### 1.1. Spherical Crystals in the Prismatine Rock from Waldheim/Saxony

The diameter of spherical to subspherical mineral grains ranges from 10 to 50 µm. The surfaces of these grains are generally very smooth [5]. Per the rock volume, the frequency of these grains amounts to about 180 spheres/cm$^3$. We used 300 µm thick doubly polished thick sections for the analyses because the volume of standard rock thin sections (~25 µm) is insufficient for the analysis of an adequate number of inclusions. We observed the release of smooth spherical crystals from the sample surface, which is due to the reduced adhesion.

The following spherical, elliptical, or subspherical crystals were identified: zircon, zircon-reidite, tourmaline, garnet, quartz, rutile, corundum, kyanite, feldspars, molybdenite, moissanite, and diamond. In prismatine, coesite forms round spherical aggregates consisting of smaller spherical coesite crystals.

### 1.2. Spherical Crystals in Topaz from the Greifenstein Granite

Based on the assumption that the formation of such supercritical fluids is a general process on Earth, one should find evidence of such fluids in other locations. Based on our observation of stishovite and coesite [1–4] in prismatine-bearing granulite rock from Waldheim, Saxony, our aim was to detect further evidence for such processes in different geological settings. Therefore, we analyzed the Variscan mineralization in the Erzgebirge. The results of numerous studies over the last 50 years demonstrated the systematic distribution of boron in granite (an increase in the boron concentration from E to W). Furthermore, granites in tin deposits often contain a considerable amount of carbonaceous material. Another unusual characteristic is the high hydrogen concentration detected in melt inclusions in the Ehrenfriedersdorf pegmatites [10]. Therefore, we examined the Variscan Greifenstein granite samples from a location close to the famous Ehrenfriedersdorf tin deposit in Erzgebirge, Germany, for supercritical fluid indications (Figure 1).

We identified spherical and subspherical crystals of quartz, cristobalite, moissanite with diamond, lonsdaleite and kumdykolite, zircon, and zircon-reidite in OH-dominated topaz from the granite (Figures 2 and 3). We also found spherical crystals, mainly graphite disks, in some feldspar crystals from the same rock.

### 1.3. Sample

The first author collected the used rock sample from the Greifenstein granite cliff in the summer of 1987.

The sample is a reddish, fine-grained granite (phase B) containing ~2% topaz [12]. The modal composition of this granite (tin-bearing granite sensu stricto) is according to [12]: quartz 33.4, plagioclase 25, K-feldspar 33.6, dark mica 5.6, white mica (0.4), topaz 2, accessories 0.1. In contrast to pegmatitic and hydrothermal mineralizations related to the Variscan tin deposit, topaz from the Greifenstein granite scarcely contains any fluid and melt inclusions. Using melt inclusions in quartz and topaz, it was possible to determine the trapping conditions for these granites (four phases): 725–562 °C and 2.1–1 kbar [13]. However, numerous solid mineral inclusions, which often are spherical and disk-like, were observed. In each case, the inclusions in question are foreign minerals, primarily spherical inclusions (no melt inclusions or nanogranitoids), which are not any primary component of the hosting granite (see, for example, [14]). Noteworthy is also that all such studied inclusions are in the scale of a few micrometers (~10 µm or more) below the surface.

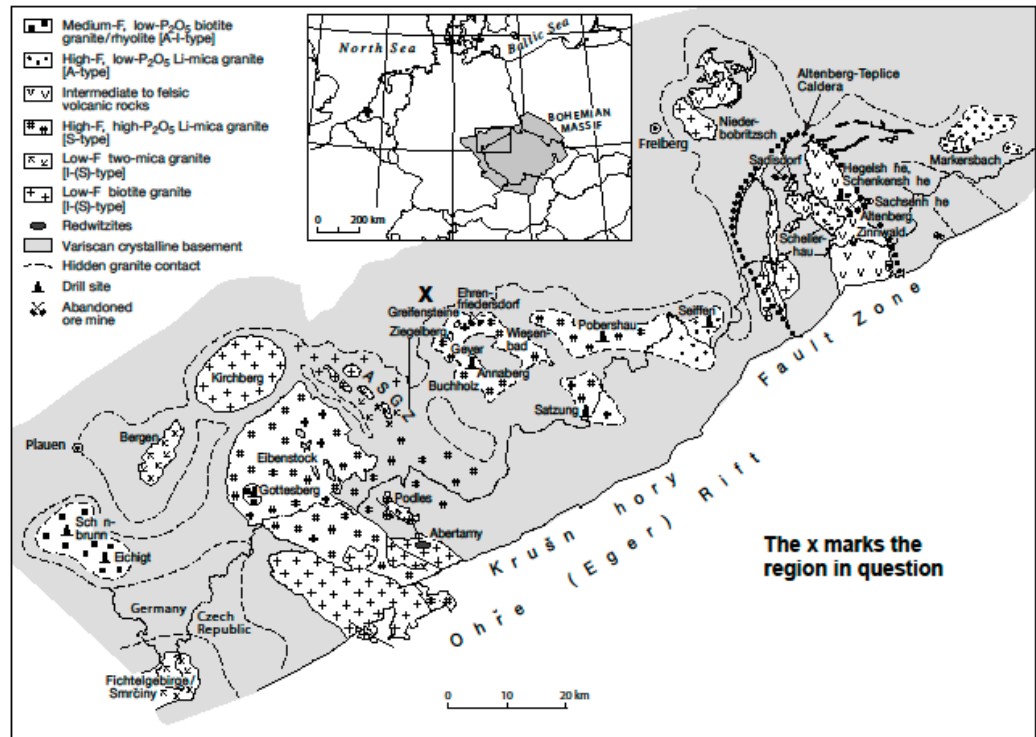

**Figure 1.** Schematic geological map of the Erzgebirge-Vogtland Zone showing the major groups and occurrences of Variscan granites, rhyolites, and redwitzites (simplified after [11] by deleting the detailed insertion map for the Aue-Schwarzenberg Granite Zone (ASGZ)). The Greifenstein granite cliff (X) has the coordinates: 50°38′57.49″ N and 12°55′48.25″ E.

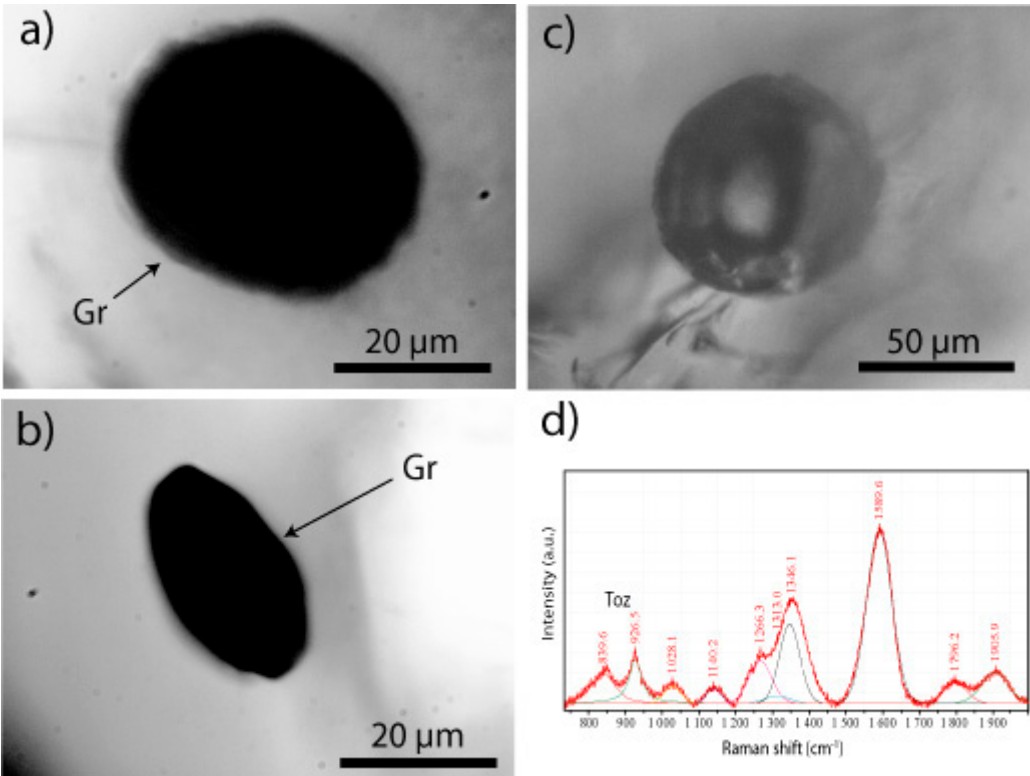

**Figure 2.** (**a**–**c**). Typical disk-like spherical graphite (Gr) inclusions in topaz from the Greifenstein granite. (**d**) Raman spectrum of the graphite inclusion shown in (**c**).

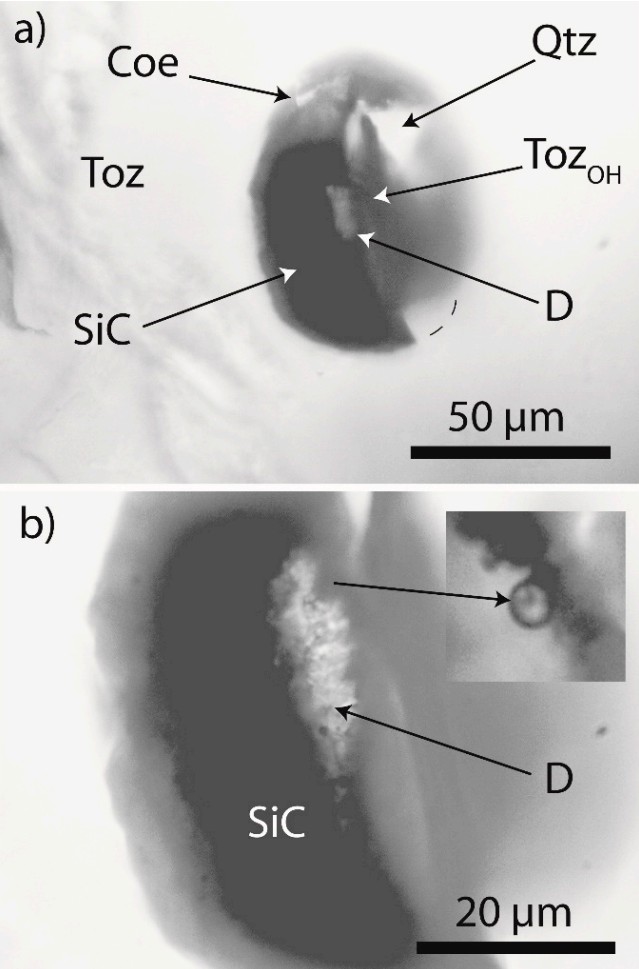

**Figure 3.** Diamond in a moissanite (SiC)-rich spherical inclusion in topaz from the Greifenstein granite (sample I). This inclusion is ~30–55 μm below the surface of the 300 μm thick sample. (**a**) Overall view. Toz—topaz host crystal, SiC—moissanite, D—diamond, and Qtz—quartz (primary stishovite or coesite; see the asymmetric volume change), Coe—coesite. (**b**) Magnified view of the diamond region shown in (**a**). The inset shows a lonsdaleite globule in diamond.

Based on the results of preliminary Raman studies (937/3650 cm$^{-1}$ band intensity), the topaz is OH-rich (OH ≥ 0.7). In addition, several spherical topaz inclusions in this matrix topaz are even more enriched in OH. Disk-like inclusions of poorly crystallized graphite are relatively abundant in this topaz and feldspar (Figure 2a,b). In addition to the dominant OH-rich topaz, corroded spherical F-rich topaz grains with OH stretching modes at 3650 cm$^{-1}$ were identified. According to our microprobe results, granitic topaz is primarily an F-rich topaz [$Al_2SiO_4(F_{1,97}(OH)_{0.03})$], which was replaced by OH-rich topaz. According to Xue et al. [15], OH-rich topaz belongs to type topaz-OH I. However, we also observed the following characteristics, which are typical for topaz-OH II: a broad O-H stretching band centered at ~3500 cm$^{-1}$ (3350–3525 cm$^{-1}$) and a wide OH band between 650 and 900 cm$^{-1}$.

The following spherical–subspherical crystals were identified in rock-forming topaz: zircon, reidite-rich zircon, moissanite with diamond, graphite, jahnsite-(CaMnFe) [$CaMn^{2+}Fe^{2+}{}_2Fe^{3+}{}_2(PO_4)_4(OH)_2 \cdot 8H_2O$], caracolite [$Na_3Pb_2(SO_4)_3Cl$] (Figure 4), and colorless orthorhombic cassiterite. For example, a moissanite crystal with a size of ~115 × 70 μm and rounded corners was identified in topaz (Toz-2), which contains microcrystals of diamond, lonsdaleite, topaz (Toz-1), stishovite, and coesite. Toz-1 and Toz-2 represent F- and OH-rich topaz, respectively.

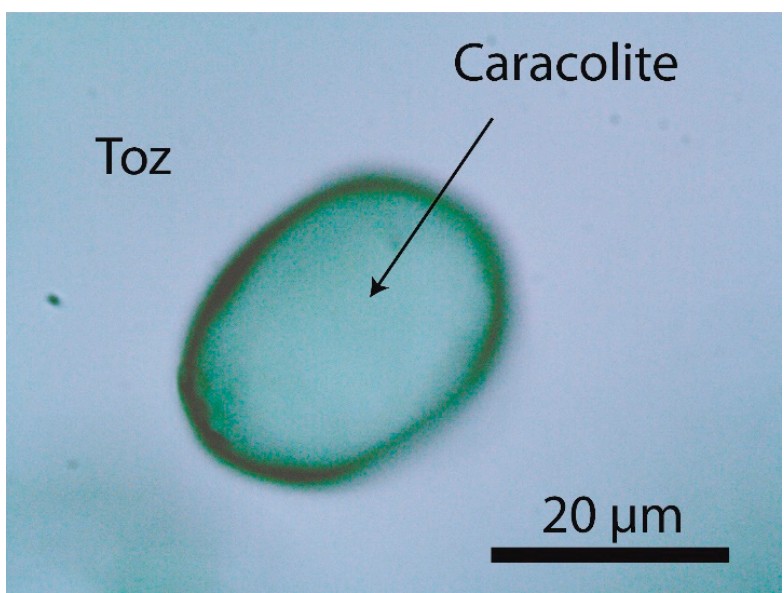

**Figure 4.** Elliptical caracolite [Na$_3$Pb$_2$(SO$_4$)$_3$Cl] inclusion in topaz (Toz-2) from the Greifenstein granite.

## 2. Results

### 2.1. Diamond and Moissanite in Granite from Greifenstein

Based on previous conclusions [16], the preservation of microdiamonds in granitic rock is extremely unlikely because of the inversion to low-pressure forms during regular transport and emplacement. Therefore, diamond and diamond-bearing moissanite differ from those in Greifenstein granite samples. To eliminate the possibility that diamond and moissanite (SiC) were introduced during cutting and grinding, we also analyzed the materials used for sample preparation (diamond cutting disc, SiC, and diamond polishing powder). The results obtained for natural and technical diamonds are provided in Table 1.

**Table 1.** Comparison of the Raman data for the first-order peak of sp$^3$-bonded carbon in natural diamond (Greifenstein granite) and that used for sample preparation.

| Sample | Band~1332 (cm$^{-1}$) | $\pm\sigma$ (cm$^{-1}$) | FWHM (cm$^{-1}$) | $\pm\sigma$ (cm$^{-1}$) | Number of Measurements |
|---|---|---|---|---|---|
| Natural diamond: | | | | | |
| Diamond in SiC-I | 1325.4 | 4.40 | 10.8 | 4.1 | 28 |
| Diamond in SiC-II | 1324.5 | 4.30 | 7.7 | 3.6 | 50 |
| Diamond in SiC-II xxx-1 | 1320.9 | 0.40 | 7.3 | 0.4 | 10 |
| Diamond in SiC-II xxx-2 | 1322.4 | 2.40 | 8.9 | 2.9 | 13 |
| Diamond in SiC-III | 1322.8 | 2.80 | 9.4 | 5.0 | 20 |
| Diamond in SiC | 1323.9 | 3.55 | 8.8 | 3.6 | 121 |
| Diamond in cristobalite | 1331.1 | 1.10 | 6.7 | 1.3 | 10 |
| Diamond in MI in zircon * | 1330.5 | 0.55 | 5.8 | 1.2 | 41 |
| Technical diamond for sample preparation: | | | | | |
| Diamond cutting disk [1] | 1332.5 | 0.42 | 5.5 | 0.3 | 28 |
| Diamond cutting disk [2],[4] | 1332.5 | 0.53 | 4.9 | 0.14 | 28 |
| Polishing wheel [3],[4] | 1330.5 | 5.40 | 31.0 | 16.6 | 28 |
| Diamond spray D1 [5] | 1336.4 | 6.90 | 67.8 | 17.0 | 28 |

**Table 1.** *Cont.*

| Sample | Band~1332 (cm$^{-1}$) | ±σ (cm$^{-1}$) | FWHM (cm$^{-1}$) | ±σ (cm$^{-1}$) | Number of Measurements |
|---|---|---|---|---|---|
| Diamond spray D0.25 [6] | 1334.6 | 5.80 | 100.0 | 6.5 | 28 |

Notes: [1] Diamond cutting disc (D30-G No. 4,716,863 Diamant Boart, Belgium: 30 μm); [2] Diamond cutting disc (A380/63, 100 μm, Russia, No. 918); [3] Polishing wheel D3, 3 μm, very heterogeneous; [4] Not used for the sample preparation; [5] DP-Spray P D1, SPRON, Struers A/S Peterstrupvej 84, DK.2750 Ballerup, Denmark; [6] DP-Spray P D0.25, SPRON, Struers A/S Peterstrupvej 84, DK.2750 Ballerup, Denmark. * Diamond in melt inclusions in zircon from UHP metamorphic rocks of the Saidenbach reservoir; unpublished data from J. Rötzler and R. Thomas (2013). In the inclusion of this rock, the first author has also determined coesite with the very strong band at 521 cm$^{-1}$ and the strong bands at 149.2 and 269.0 cm$^{-1}$.

Our data indicate: (1) a significant difference between the bivariate average values of natural diamond from the Greifenstein granite and diamond cutting discs used for the sample preparation (Diamant Boart, Belgium), at a statistical certainty of 0.999; and (2) all diamonds used for cutting and polishing are cubic, whereas natural diamond from the Greifenstein granite is hexagonal.

The diamond cutting disc (D30-G No. 4,716,863 Diamant Boart, Belgium) has a mean grain size of 30 μm. The Raman spectra of diamonds from the cutting disk exhibit a peak at $1332.5 \pm 0.4$ cm$^{-1}$ with a full width at half maximum (FWHM) of $5.5 \pm 0.3$ cm$^{-1}$ (28 measurements), representative of a well-crystallized cubic diamond. The Raman spectra of the 1 μm DP-Spray P diamond polishing powder (SPRON, Struers A/S Peterstrupvej 84, DK-2750 Ballerup) are characterized by a peak at $1336.4 \pm 6.9$ cm$^{-1}$ and FWHM of $67.8 \pm 17.0$ cm$^{-1}$ (28 measurements). It should be noted that we obtained poor Raman spectra for the ¼ μm DP-Spray P ($1334.6 \pm 5.80$ cm$^{-1}$; FWHM = $100 \pm 6.5$ cm$^{-1}$, n = 28; strong G-band at 1580 cm$^{-1}$, FWHM = 75 cm$^{-1}$).

*2.2. Natural Diamond and Moissanite in Inclusions in Granite*

During the microscopic study of two doubly polished thick sections ($3.5 \times 2.0$ cm; ~300 μm thick) and several small polished chips from the Greifenstein granite, we identified spherical aggregates of moissanite (SiC) in topaz crystals, some of which contain microdiamonds. Figure 2 shows a large aggregate with an elliptical cross-section of $74 \times 55$ μm. The largest area of diamond in this moissanite in the Greifenstein topaz is ~$25 \times 7$ μm (sample I). Figure 5 shows another slightly larger aggregate ($118 \times 73$ μm; sample II). This aggregate is not spherical but has strongly rounded peripheries. This aggregate in OH-rich topaz (Toz-2) contains moissanite, quartz, topaz (Toz-1), and diamond, as well as smaller diamond grains and microcrystals of coesite and metastable stishovite. The shape and composition indicate the natural origin of this complex SiC aggregate. Such an aggregate cannot be introduced during cutting and grinding. Note that the SiC powder has a grain size of 10 μm and exhibits sharp peripheries.

The diamonds in moissanite in topaz from the Greifenstein granite (Figure 3) mainly belong to the hexagonal diamond polytype (Figure 6). Based on 28 measurements, we obtained a value of $1325.4 \pm 4.4$ cm$^{-1}$ for the main Raman peak of the diamond. The FWHM is $10.8 \pm 4.1$ cm$^{-1}$. The value for the diamonds in the second moissanite sample was determined to be $1320.9 \pm 0.4$ cm$^{-1}$ (diamond xxx-1). In contrast to diamond polishing powder (D1 and D0.25 μm), Raman spectra of natural diamonds exhibit sharp lines. The Raman spectra of several diamond grains correspond to those of natural lonsdaleite (Figures 6 and 7) [17].

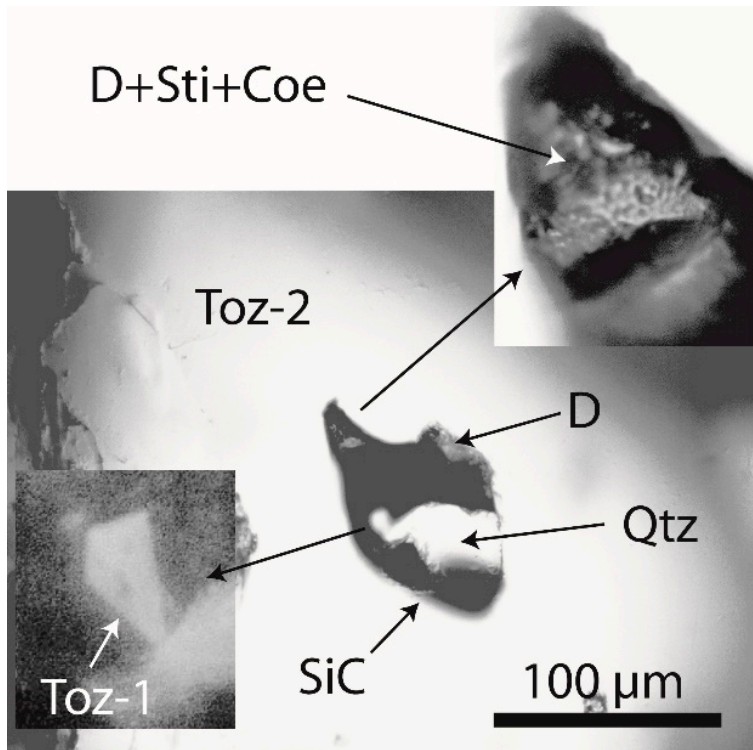

**Figure 5.** Complex moissanite (SiC) inclusion in topaz (Toz-2) from the Greifenstein granite. D—diamond, Sti—stishovite, Coe—coesite, Qtz—quartz. Toz-1 is a high-pressure topaz in the moissanite aggregate. The paragenesis confirms that the moissanite inclusion was not introduced during the sample preparation.

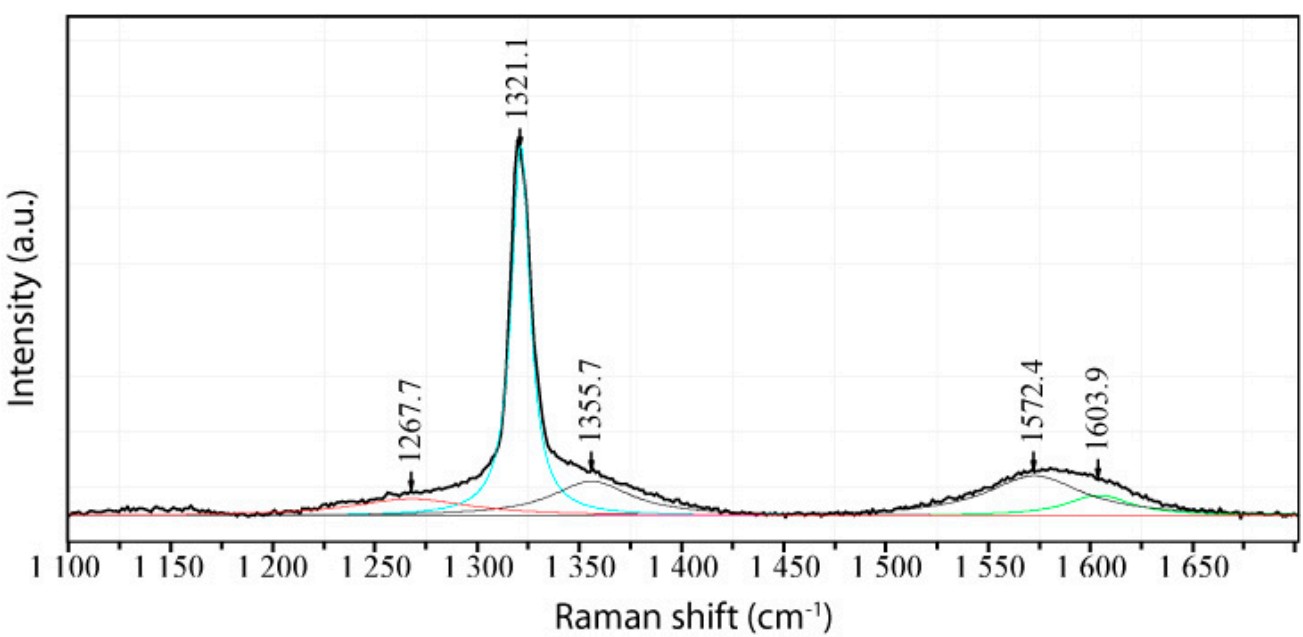

**Figure 6.** Raman spectrum of hexagonal diamond in the large diamond area shown in Figure 3.

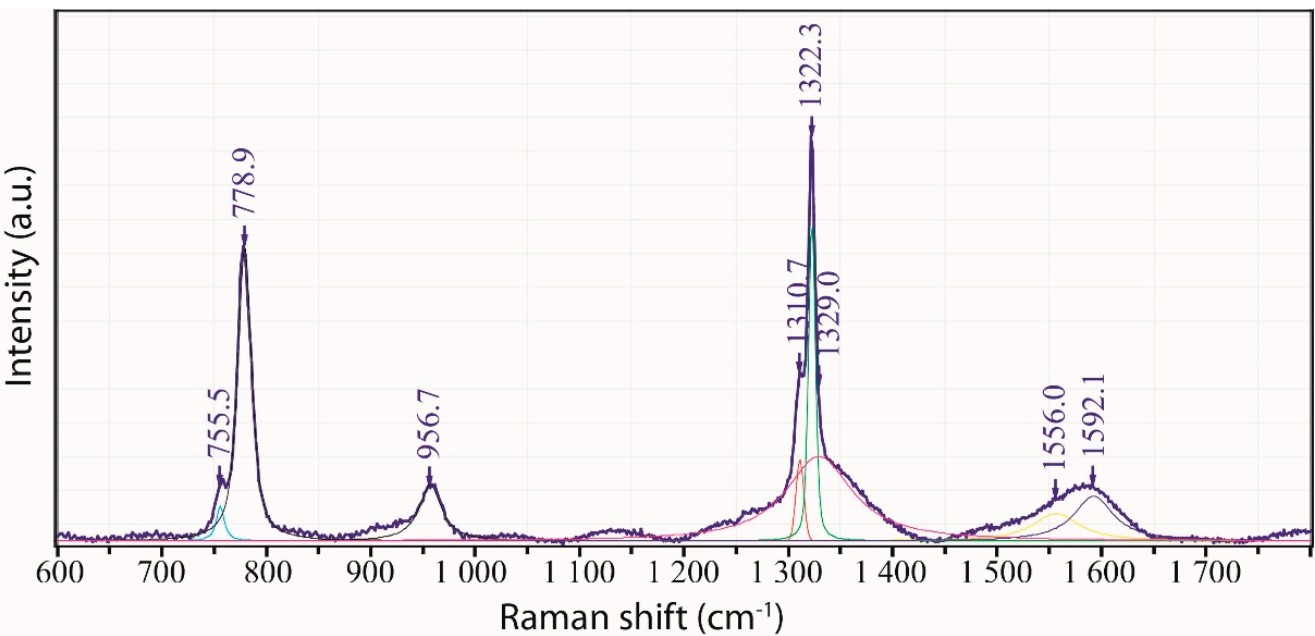

**Figure 7.** Raman spectrum of moissanite (bands at 755.5, 778.9, and 956.7 cm$^{-1}$) containing two-phase particles of diamond-lonsdaleite (bands at 1310.7, 1322.3, and 1329.0 cm$^{-1}$). Based on previous results [18], the location of the sp$^3$ breathing vibration mode of lonsdaleite, that is, hexagonal diamond, varies from 1320 to 1327 cm$^{-1}$.

Data for the natural diamond are summarized in Table 1. Remnants of coesite also occur in the moissanite–diamond–quartz–topaz inclusion (Figures 3 and 5). This is significant evidence for a high-pressure origin because moissanite, diamond, and coesite are all above their stability fields at the formation depth of the Greifenstein granite. All three phases co-occur in crystal aggregates found as inclusions in topaz. Based on the Raman spectra of coesite (sample I), a mean of 519 ± 0.7 cm$^{-1}$ was obtained for the position of the main coesite main. The following rather weak bands can also be used for the identification: 76, 153, 357, and 785 cm$^{-1}$. Based on 16 measurements of the symmetric stretching mode [19] in coesite from sample II, the mean value of the main coesite band is $\nu_s = 518.8 \pm 4.4$ cm$^{-1}$ (FWHM = 11.2 ± 5.2 cm$^{-1}$). The following values (cm$^{-1}$) were obtained for the other diagnostic bands: 76.2 ± 0.7, 153.6 ± 0.2, 331.4 ± 0.4, 787.4 ± 1.9, and 1163.7 ± 3.4. In addition to coesite, rare and metastable stishovite was observed. The following values were obtained from the analyses of four stishovite grains: 760.8 ± 1.8, 586.8 ± 2.8, and 232.2 ± 2.4 cm$^{-1}$.

A diamond crystal (~20 × 15) μm in sample II yields a value of 1320.9 ± 0.4 cm$^{-1}$ with an FWHM of 7.3 ± 0.4 cm$^{-1}$ (n = 10), corresponding to the hexagonal diamond polytype lonsdaleite [17,20]. This type of diamond is not used for sample preparation.

Based on ten measurements, we obtained a value of 1331.1 ± 1.1 cm$^{-1}$ (FWHM = 6.7 ± 1.3 cm$^{-1}$) for the diamond in the corroded cristobalite crystal (Figure 8). All diamond grains from the Greifenstein granite show a prominent G band at ~1590 cm$^{-1}$ of carbonaceous material.

In addition to diamond, the small elliptical moissanite inclusion in topaz (sample III) shown in Figure 9 contains an aggregate of cristobalite with tiny coesite crystals, cristobalite-II, or Si with a very strong dominant Raman band at 518.2 ± 0.8 cm$^{-1}$ and FWHM = 6.4 ± 1.3 cm$^{-1}$ (n = 24). Note that a clear differentiation between the three minerals is impossible because the strong and broad Raman bands of cristobalite and moissanite overlap. Černok [21] reported that cristobalite-II has a strong band at ~519 cm$^{-1}$ (at 2 GPa), which competes with the main coesite band. The intense and sharp Raman band at 713.7 ± 1.1 cm$^{-1}$ (FWHM = 13.4 ± 1.4 cm$^{-1}$) indicates the presence of cristobalite-X-I.

Further research is necessary in the near future because the proof of cristobalite-X-I would indicate a significantly higher pressure (~10 GPa and more).

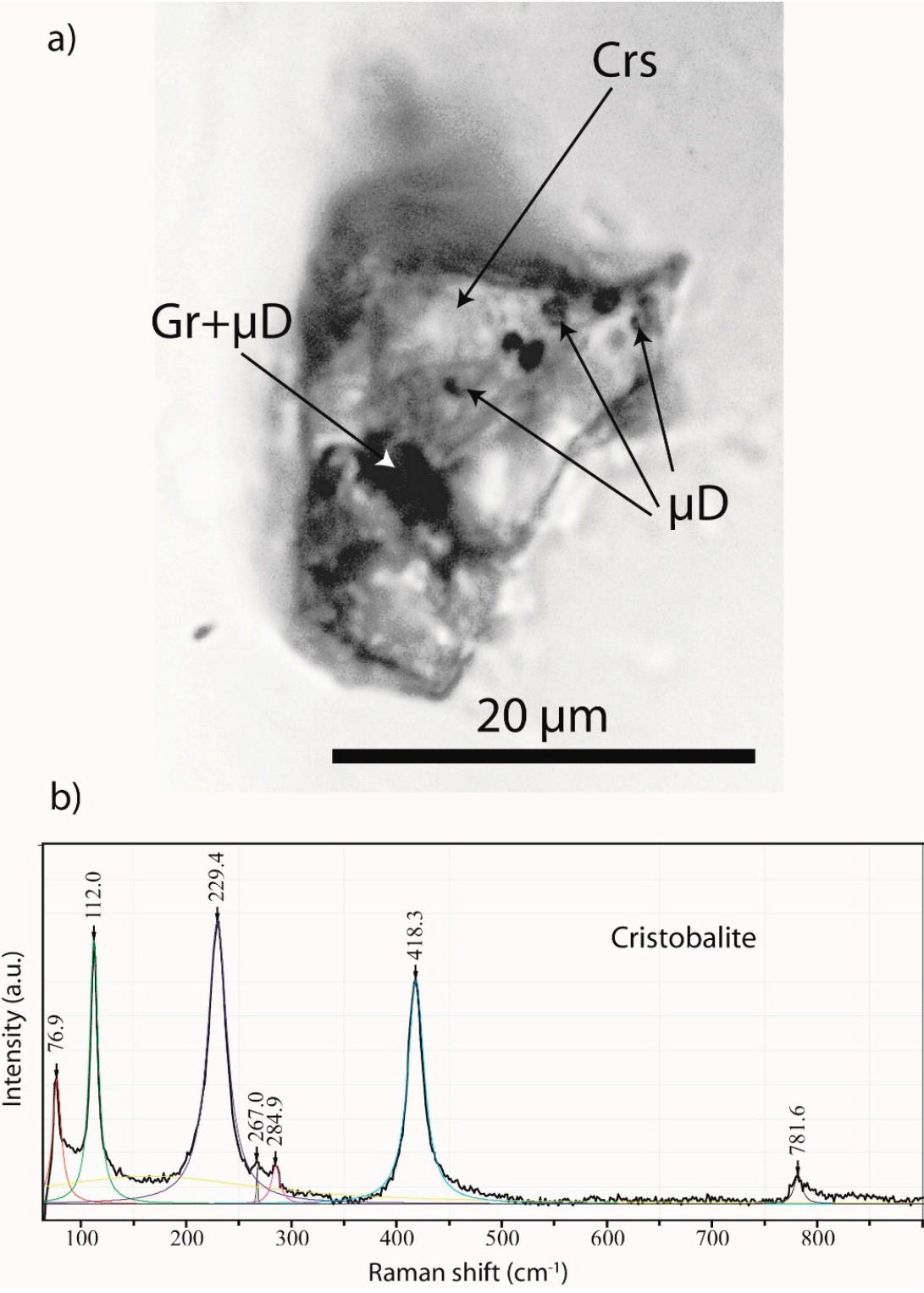

**Figure 8.** Corroded cristobalite (Crs) crystal with graphite (Gr) and microdiamond (μD) in topaz (**a**). Cristobalite is the stable SiO$_2$ form that forms from stishovite or coesite, which are metastable under granite crystallization conditions. (**b**) Raman spectrum of cristobalite shown in (**a**).

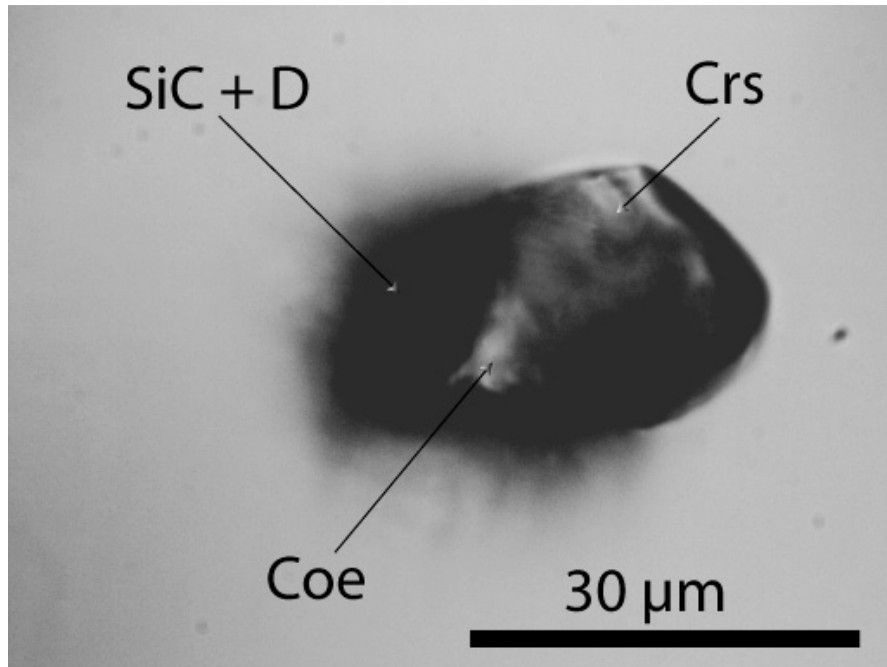

**Figure 9.** A small elliptical moissanite inclusion in topaz (sample III) contains an aggregate of α-cristobalite with tiny diamond and coesite crystals, cristobalite-II, and maybe cristobalite-X-I.

The spectrum of the moissanite crystal [SiC] in the inclusion (Figure 2) of sample I is characterized by strong lines at $774.4 \pm 7.2$ cm$^{-1}$ (FWHM = 24.5 cm$^{-1}$) and $945.1 \pm 12.0$ cm$^{-1}$ (FWHM = 24.2 cm$^{-1}$). This crystal is significantly larger ($67 \times 27$ μm) than the grains in SiC powder used for grinding (F10 with a 10 μm mean grain size). This technical-grade SiC is of type 6H-SiC and exhibits prominent bands at $784.0 \pm 3.6$ cm$^{-1}$ and $963.1 \pm 2.4$ cm$^{-1}$ with an FWHM of $7.1 \pm 1.7$ cm$^{-1}$ and $19.8 \pm 14.0$ cm$^{-1}$, respectively (15 different grains). Natural and technical SiC significantly differ, not only in size.

Based on the size, spherical form, and diamond type, diamond, lonsdaleite, graphite, moissanite, and kumdykolite do not represent contamination of topaz from the Greifenstein granite that was introduced during sample preparation. We propose that they represent phases that are most likely rapidly transported by a supercritical fluid from a depth corresponding to a pressure of ≥5 GPa or, in the case of cristobalite X-I of ~11 GPa, to the location of granite crystallization at about 0.21 to 0.1 GPa [13] in analogy to coesite in prismatine from the Waldheim granulite trapped at about 1.3 GPa at 1000 °C [4].

Based on the assumption that diamond, lonsdaleite, moissanite, graphite, and kumdykolite were transported by a supercritical fluid, energy, volatiles, and a broad spectrum of main and trace elements would also be added to the crystallizing granite. This indicates the necessity of a reassessment of the genetic concept of element enrichment. In addition, the generally accepted origin of the topaz from the Greifenstein granite as a late phase must be scrutinized because the diamond, graphite, SiC, cristobalite, and coesite inclusions in topaz imply a different formation of this topaz type. Note that such hydroxyl-rich topaz has been identified in ultrahigh-pressure metamorphic rocks [22].

## 3. Discussion

The results of experiments showed that water and aluminosilicate melt are miscible in any proportion at pressures > 3 GPa [23,24]. We propose that aliquots of such a melt with very high water concentrations would have an extremely low density and viscosity as well as an enormous capacity to sequester volatile elements such as boron. This type of fluid would be highly mobile and very buoyant at such depths. However, under favorable conditions, it may be able to entrain tiny crystals present at these depths. This type of fluid would also be highly fugitive at upper crustal levels and would tend to dissociate

into volatile-rich fluids, aluminosilicate melts, and crystals entrained during its ascent. In previous publications [25], we proposed that such fluids can explain the unusual chemistry of pegmatites and other characteristics. The presence of stishovite and coesite inclusions suggests that similar fluids may originate at much greater depths than anticipated. In the past [26], we referred to these fluids as supercritical fluids. However, these fluids may correspond to previously described "transcrustal fluids" [27] because of their extraordinary physicochemical properties. They differ from but may be related to kimberlite magmas, which can also transport diamonds from the ultra-deep crust to the surface rapidly enough to prevent the inversion to graphite [28].

The rapid injection of boron, water, and its crystal load into the crustal region in Waldheim provides favorable conditions for the pegmatite-type crystallization of prismatine rocks, at least locally. We refer to this type of crystallization as pegmatite crystallization induced by supercritical fluids because it slightly differs from the typical formation of granitic pegmatites. The significance of supercritical fluids for the formation of pegmatites and ore deposits must be investigated in the future.

The origin of the inclusions trapped by moissanite in topaz remains unclear. For example, the data obtained for diamond (Table 1) (except for the Saidenbachtal data) plot on the left edge and lower corner of Figure 6 in a recent study [29]. Only a few data points plot in this field. Greifenstein UHP minerals represent a new type. The diamonds significantly differ from those described in ultrahigh-pressure metamorphic rocks from the Saidenbachtal reservoir composed of garnet–phengite gneiss close to the Greifenstein granite [29–31] and Table 1. Therefore, further in-depth studies should be carried out in the future (e.g., analysis of C-isotopes of diamond and moissanite in addition to transmission electron microscopy TEM).

## 4. Conclusions

The discovery of co-trapped inclusions of ultrahigh-pressure minerals, including diamond, moissanite, coesite, stishovite, lonsdaleite, kumdykolite, and reidite, in minerals in crustal rocks provides evidence for their trapping at much lower pressures. The explanation of this phenomenon requires an inclusion transport mechanism that is much more rapid than that typically associated with transport from lower crustal depths, that is, a mechanism that is fast enough to prevent the inversion of high-pressure minerals into their lower-pressure polymorphs. We propose that melt/fluids in which $H_2O$ and aluminosilicate melt are miscible in any proportion (in our terminology, supercritical fluids) may provide the carrier for such tiny crystals of high- and ultrahigh-pressure minerals. Most likely, such fluids may also carry significant concentrations of other volatiles and metals. We previously showed that these supercritical fluids could dissolve much higher concentrations of a wide range of elements than conventional hydrothermal fluids [25]. Although extremely fugitive at upper crustal pressures, such processes may at least partially prove why not all granites are metal-rich or produce ore deposits. Furthermore, at the moment, very speculative, it is quite conceivable that an old subducted tin deposit is partially activated. Such a synergetic process can explain this deposit's extraordinary richness and variety. Here we mean the extreme richness in simple and evolved pegmatites [32].

We recommend that a diamond or SiC embedded in the surface of a polished section of granite should not automatically be interpreted as contamination and ignored, but in-depth analyses should be carried out. Firstly, are the diamonds and SiC the same as those used for sample preparation? Raman analysis can be used to determine properties beyond simple composition, and significant discrepancies may indicate a different origin. In addition, disk-like to spherical, very smooth solid inclusions (transparent or opaque) enclosed in other minerals formed at high temperatures without any sign of equilibrium and recrystallization should be closely examined. Such inclusions indicate co-trapping during host mineral crystallization. Raman microanalysis is ideal for the analysis of such inclusions because it is non-destructive and allows for precise crystal structure determination, in contrast to other

analytical methods. Therefore, Raman spectroscopy is a precious tool for the exploration of important "nearly invisible" evidence of hidden fundamental processes.

Sensitized by the identification of high-pressure signatures in prismatine rock from Waldheim and the Greifenstein tin-bearing granite, we searched for and found similar spherical crystals in other rocks (Cadomian granodiorites and quartz veins of the Lausitz Block/E-Germany, young granites from Eldzhurtinsk/Caucasus), demonstrating an interaction between mantle and crust.

**Author Contributions:** R.T., P.D., A.R. and U.R. designed the project; R.T. performed the Raman analyses. All authors have read and agreed to the published version of the manuscript.

**Funding:** This research received no external funding.

**Institutional Review Board Statement:** Not applicable.

**Informed Consent Statement:** Not applicable.

**Data Availability Statement:** Not applicable.

**Acknowledgments:** Late Otto Leeder (Freiberg) and Günter Meinel (Jena) are thanked for sensitizing the first author concerning observations contradicting mainstream ideas. We thank Sylvia-Monique Thomas (Las Vegas) for editing the English text. We are also indebted to A. Acosta-Vigil, as well as to three anonymous reviewers for the pertinent comments on this manuscript. For the reprint permission of the map (Figure 1) on the DVD to the reference [12], we thank the E. Schweizerbart'sche Verlagsbuchhandlung Stuttgart, Germany. The authors thank Jochen Rötzler for making available unpublished data on the Saidenbach reservoir.

**Conflicts of Interest:** The authors declare no conflict of interest.

## Appendix A

*Microscopy and Raman Spectroscopy: Methodology*

We have performed all microscopic studies with a petrographic polarization microscope coupled with the Raman spectrometer. The Raman spectra were recorded with an EnSpectr Raman microscope RamMics M532 in the spectral range of 0–4000 cm$^{-1}$ using a 50 mW single-mode 532 nm laser, entrance aperture of 20 μm, holographic grating of 1800 g/mm, and spectral resolution ranging from 4–6 cm$^{-1}$. Depending on the grain size, we used microscope objective lenses with a magnification varying from 3.2× to 100×. We used the Olympus long-distance LMPLFLN100x as a 100x objective lens. The laser energy on the sample can be adjusted down to 0.02 mW. The Raman band positions were calibrated before and after each series of measurements using the Si band of a semiconductor-grade silicon single-crystal. The run-to-run repeatability of the line position (based on 20 measurements each) was ±0.3 cm$^{-1}$ for Si (520.4 ± 0.3 cm$^{-1}$) and 0.5 cm$^{-1}$ for diamond (1332.3 ± 0.5 cm$^{-1}$ over the range of 80–2000 cm$^{-1}$). We used a water-free natural diamond crystal as a diamond reference (for more information, see [4]).

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
