# Peer review of "Ultrahigh-Pressure Mineral Inclusions in a Crustal Granite: Evidence for a Novel Transcrustal Transport Mechanism"

_geosciences, doi:10.3390/geosciences13040094_

Round 1
Reviewer 1 Report (New Reviewer)
The article considers the mechanism of formation of minerals at ultra-high pressure in intrusive granites on the example of stishovite and coesite.
From the standpoint of an accelerated transport mechanism, the presence of a group of minerals in topazes of granite rocks was established, which confirms the concept of direct interaction between the mantle and the crust through fluids of volatile substances.
Systematized data on the presence of inclusions of ultra-high pressure minerals in the rocks of the crust, indicating the occurrence of mineralization processes at lower pressure.
The authors proposed the concept that liquid melts hold crystals of high and ultra-high pressure minerals with a higher concentration of elements than simple hydrotherms. This explains why ore deposits in granites are not always formed. To determine the properties of minerals in such cases, a method of combination analysis is proposed. The carrier of information about the genesis of the inclusion is the particles formed at high temperatures without recrystallization and indicating co-trapping during crystallization.
The advantage of the method is that it is non-destructive and more accurate, so it can capture the details of the hidden processes of mineralization.
The authors found confirmation of their hypothesis in the rocks of the Waldheim, Greifenstein, Lausitz (Germany), and Tyrnyauza (Russia) deposits
With a general positive assessment of the article, a private remark suggests itself. Since the ultimate goal of the article is, after all, to substantiate the expediency of using this area of the earth's crust for mining, it would be good to outline the contours of the methodology for predicting an increase in the reserves of a deposit due to the phenomenon of mineralization protected by the authors.
Author Response
Dear Sir, Dear Madam,
Thank you very much for the reviews of the manuscript ID: geosciences-2241119. It isn't straightforward for me to revise the manuscript at the moment, and by my stroke, I am a little bit handicapped. My coauthors are in a similar situation. Therefore, I excuse any delay in the handling of the manuscript.
Response for Reviewer 1:
The reviewer has not underlined specific points. Predicting the increase in the reserve of a deposit is the second step. First, we must bring the proof for a completely new view of an ancient mining district.
Thank you and best regards,
Rainer Thomas
Reviewer 2 Report (New Reviewer)
Incredulous, so good luck.
I have no faith in ruling out all contamination from preparation.
Only if you prepared the samples yourself by hand, would I put my name on it.
Best of Luck though.
Author Response
Dear Sir, Dear Madam,
Thank you very much for the reviews of the manuscript ID: geosciences-2241119. It isn't straightforward for me to revise the manuscript at the moment, and by my stroke, I am a little bit handicapped. My coauthors are in a similar situation. Therefore, I excuse any delay in the handling of the manuscript.
Response for Reviewer 2:
The reviewer has not underlined specific points. Yes, I have taken the sample and prepared these in part.
I hope I have improved the manuscript according to the reviewers significantly, and it can now be used for publication.
Thank you and best regards,
Rainer Thomas
Reviewer 3 Report (New Reviewer)
Lines 2-3, Title: Granites of all types are intrusive rocks either they form from melts in the crust or melts from the mantle, and accordingly granites are mostly a crustal lithology and much less abundant plagiogranite, which form from mantle-derived magma. Therefore, it is recommended here to modify your title from:
Ultrahigh-pressure Mineral Inclusions in an intrusive Granitic Rock: Evidence for a Novel Transcrustal Transport Mechanism TO Ultrahigh-pressure Mineral Inclusions in Crustal Granite: Evidence for a Novel Transcrustal Transport Mechanism.
Line 11, Highlights: Your samples are taken from a tin-bearing granite from a single locality (the Erzgiberge, Germany), and from different locations of the same granitic volume. So, try to avoid using “different locations”.
Lines 15-25, Abstract: Only minor corrections are needed as indicated in the attached pdf.
Line 26, Keywords: I think you need to include UHP minerals among your keywords.
Line 31, Introduction: As far as data in the provided appendix are not huge, so it is better to delete and move the information it includes to the methodology section.
Line 40: Is the range of diameter is that wide?!. I think you mean (size range 10-50 μm).
Line 42: Dravite is a variety of tourmaline so it is better to use “tourmaline” instead.
Line 48: Use "sphere-in-sphere" instead of "spheres within spheres".
Line 51: If you consider your spherical inclusions "foreign bodies", then you need to use "exogenous" instead of "extraneous" in line 4.
You need to separate materials and results from the introduction section. It is better to include the sentences about the materials used (UHP minerals and their host spheres), side by side with methodology in Appendix 1. Then, use the rest of sentences in the results section that follows methodology.
Lines 70-75: This part represents is good for the aims of study, alongside with the following sentences, it can be formulated as a separate short paragraph.
Line 93, Fig. 1: Legend of the map is pixel. Font should be at least 300 dpi for reproduction I guess. Also, latitudes, longitudes and geographic north of the map are missing. Any geological map, even a sketch or simplified one, should include all of them.
Line 95: What kind of modifications have been made by the present authors for the map drafted by [12]?.
Lines 109-110: You wrote “Granitic topaz is primarily F-rich topaz, which was replaced by OH-rich topaz”. Do you have any petrographical evidence for this observation, which is time-wise a conclusion too?.
Line 128: At the end of the paragraph, you need to say: Toz-1 and Toz-2 reprsenet F- and Oh-bearing topaz, respectively.
Line 130, Fig. 4: Which topaz (1 or 2) is the host for the caracolite inclusion?.
Lines 134-135: Regular transport, do you mean emplacement or move to shallower level in the earth’s crust?.
Lines 229-230, Fig. 9: Yu mentioned about cristobalite-X. You can delete this from the figure caption and keep it in the text.
Lines 236-237: Here you consider the technical SiC as synthetic, which is correct and you need to unify all over the text.
Lines 240-242: You stated “We propose that UHP mineral inclusions represent phases that are rapidly transported by a supercritical fluid from a depth corresponding to a pressure of ≥5 GPa to the location of granite crystallization in analogy to coesite in prismatine from the Waldheim granulite. If you present such an analogue, you need to provide “conclusive” evidence about the thermodynamics of the system because crsytallization in a metamorphic environment (the case of granulite) differs from that of granite that should originate here from a very deep seated felsic melt. Otherwise, you need to give evidence that the supercritical fluids are derived from the mantle.
Lines 250-251: Again, crystallization of OH-bearing topaz during metamorphism is not identical to that forms in an igneous system, i.e. the felsic magma.
Lines 25-282, Discussion: It should be more enhanced. You need to reduce size of conclusions (Lines 283-319) and include parts of them in the discussion section. Your conclusions should be straightforward, concise and possibly in the form of bullets. This would be more informative.
Line 334, Appendix 1: It is better to include in the text as a methodology section, side by side with parts from the section before results.
Lines 349-413, References: You need to re-edit some of the references in the list following a careful usage of the journal instructions for the preparation of the reference list. The apparent issue is that you abbreviate some journals and you do not for some others. I highlighted some in the attached annotated pdf, e.g. [10[, [13], [18] and [25].

Author Response
Dear Sir, Dear Madam,
Thank you very much for the reviews of the manuscript ID: geosciences-2241119. It isn't straightforward for me to revise the manuscript at the moment, and by my stroke, I am a little bit handicapped. My coauthors are in a similar situation. Therefore, I excuse any delay in the handling of the manuscript.
Responses for Reviewer 3:
This reviewer made many good points, which I accepted nearly all of. However, I have left the methodical points in Appendix 1.
Line 2-3: I have accepted the changed title of the manuscript.
Keywords: were completed.
Line 93, 95: I have the caption for the map completed by latitude and longitude of the Greifenstein granite cliff, and shortly explained what I have done by simplification. The inserted small overview map indicates the geographic north.
Line 109-110: I have included the topaz formula determined by my microprobe analyses.
Line 130: Topaz type included.
Line 140, Table 1: I have completed table 1 by unpublished Raman data for diamonds from the Saidenbach reservoir - the data I measured in 2013 on material from Jochen Rötzler. He is thanked in the acknowledgment.
Line 226: The sentence was improved.
Line 236: For the term "synthetical," we now use "technical."
Line 240/242: This sentence was now improved.
Lines 250: Improved.
Line 335: The Acknowledgments are completed.
The style of the references [10[, [13], [18], and [25] were adapted.
I hope I have improved the manuscript according to the reviewers significantly, and it can now be used for publication.
Thank you and best regards,
Rainer Thomas
This manuscript is a resubmission of an earlier submission. The following is a list of the peer review reports and author responses from that submission.
Round 1
Reviewer 1 Report
This paper presents the description of co-trapped inclusions of very high-pressure minerals including diamond, moissanite, coesite, stishovite, lonsdaleite and reidite in minerals in crust-derived rocks. The granite rocks were formed at much lower pressures compared to that of these minerals and thus this requires a transport mechanism for those inclusions far more rapid than those typically associated with transport from lower crust or upper mantle. This is rather interesting and the authors proposed that these minerals were transported by supercritical liquids (H2O+aluminisilicate melts), which sounds reasonable. The study is informative, novel and inspiring and thus I suggest a fast publication after minor revision.
Comments:
1) Figure 1a and Figure 1b should be combined as Fig. 1.
2) If these high pressure minerals were transported by supercritical fluids, is there any interaction between such minerals and SCF?
3) "We refer to this type of crystallization as pegmatite crystallization induced by supercritical fluids, as it differs somewhat from the typical formation of granitic pegmatites." What are the differences?
I hope it helps,
Best reagrds.
Author Response
Reviewer 1
The reviewer has recommended combining Fig. 1 and Figure 1b. We have done this, and some minor revisions are in the works (see reviewers 2 and 3).
Of course, there is an interaction of the SCF (supercritical fluids). However, the sheathing of the minerals with SiC prevents such a reaction. Only by such protection, the specific minerals can survive.
The difference to the classic granite pegmatites is the absence of the free room. Most inclusions are melt inclusions, and primary fluid and CO2 inclusions are rare.
Reviewer 2 Report
The manuscript entitled “Ultra-High Pressure Mineral Inclusions in Crustal Rocks: Evidence for a Novel Trans-Crustal Transport Mechanism” by R. Thomas and co-authors reports a Raman spectroscopy study of host-inclusion systems from a Variscan tin granite from the Erzgebirge deposit. Given the detection of a suite of high-pressure minerals, such as moissanite and coesite in these rocks, the authors ascribed their unusual occurrence in crustal rocks to an interaction between mantle and crust via supercritical fluids or volatile-rich melts.
However, in my opinion the conclusions are not supported by the results and viceversa. Raman spectroscopy analyses only are not sufficient to support the authors statements and moreover the reported Raman spectra are incomplete since for some mineral phases cited in the text there is not the Raman spectrum. Also, since they are host-inclusion systems is very difficult to distinguish and identify all the phases giving peak overlaps. Moreover, about lonsdaleite there are several studies which in the last 3 years have widely discussed the literature and deepened the investigation of this phase showing that only one technique is not sufficient to describe this kind of structures. I will report some of them here below.
Furthermore, also a re-organization of the logic and a careful English check are required, otherwise is difficult to read it through.
Overall, this manuscript needs substantial work before being considered as a future publication.
Therefore, I would recommend the rejection in its actual form.
Suggested literature for the study of carbon phases:
Murri, M., Smith, R. L., McColl, K., Hart, M., Alvaro, M., Jones, A. P., ... & McMillan, P. F. (2019). Quantifying hexagonal stacking in diamond. Scientific reports, 9(1), 1-8.
Németh, P., McColl, K., Smith, R. L., Murri, M., Garvie, L. A., Alvaro, M., ... & McMillan, P. F. (2020). Diamond-graphene composite nanostructures. Nano letters, 20(5), 3611-3619.
Németh, P., McColl, K., Garvie, L. A., Salzmann, C. G., Murri, M., & McMillan, P. F. (2020). Complex nanostructures in diamond. Nature materials, 19(11), 1126-1131.
Németh, P., Lancaster, H. J., Salzmann, C. G., McColl, K., Fogarassy, Z., Garvie, L. A., ... & McMillan, P. F. (2022). Shock-formed carbon materials with intergrown sp3-and sp2-bonded nanostructured units. Proceedings of the National Academy of Sciences, 119(30), e2203672119.
Author Response
Reviewer 2
Reviewer 2 has not realized that our finding of UHP mineral is now the first one in such an intrusive granitic rock. All described transport mechanisms are different concerning our case, and Russel et al. (2012) describe a completely different scenario. Also, Stöckhert et al. (2001) speak from micro-diamond daughter crystals in garnet precipitated from supercritical COH + silicate fluids. The product is very different from our findings, and there are melt inclusions with diamonds. What is conspicuously is that a row of authors who have described the diamondiferous quartzofelspathic rocks from the Erzgebirge does not call the location (Stöckhert et al., 2001, Kotková et al. (2021). By this results in some inconsistencies.
The corrected manuscript now gives some references dealing with the diamond's different occurrences in the Erzgebirge.
In our contribution, we used as a first task the light-microscopical and Raman-spectroscopic methods for the characterization of the UHP minerals. Younger colleagues must perform further sophisticated studies because the author and coauthors have aged nearly 80 and have no access to any research facilities.
The Raman technique is adequate for first characterizing the unusual minerals. On some uncertainties (coesite – cristobalite-II), we will point.

Reviewer 3 Report
Review of: "Ultra-high pressure mineral inclusions in crustal rocks: evidence for a novel trans-crustal transport mechanism" by Rainer Thomas and co-workers
In this paper, the authors claim to have found a number of UHP minerals (diamond, moissanite, coesite, stishovite, lonsdaleite, and reidite) as inclusions in Topaz from the Geigenstein granite (Erzgebrige). While a few of these (such as diamond and coesite) have already been reported for that locality, others seem to be novel. However, the finding of phases such as stishovite, lonsdaleite, and reidite in subducted crustal rocks goes against most available tectonic models for the area. The authors do not back these claims with enough data. For this reason, a much more extensive characterization is needed with multiple analytical techniques (e.g., Raman, XRD, FT-IR, micro-tomography, etc.).
The English is poor and the overall logical structure of the paragraphs should be revised. I consider this paper not suitable for Geosciences in its present state.
Author Response
Reviewer 3
Reviewer 3 has not realized that the find of UHP minerals in topaz in the intrusive Greifenstein granite is the first one generally. The Erzgebirge is a very complex terrain, and in our contribution, we represent for the first time such minerals in topaz of a Variscan granite. This granite is related to the famous tin-tungsten deposit of Ehrenfriedersdorf. The often-described paragenesis of microdiamonds in the Erzgebirge region is associated with an ultrahigh-pressure metamorphic unit composed of garnet-phengite gneiss not far from our occurrence (~20 km as the crow flies). However, both have nothing to do with each other (at the moment).
The request for more data for an extensive characterization by multiple analytical techniques is the second step before the first. In the first step, we must show that any preparation step does not include moissanite and diamond in the described samples. Nobody will pay any penny for the case if this question is open.
The statement that the English are poor, we counter that the second author is a native speaker from Australia. Sylvia-Monique Thomas (Las Vegas) has also edited the English text.
Paul Davidson wrote to this point: I have checked your note and it seems to me that you have nailed it. However, I would not be too dismissive of the English simply because I am a native speaker. Australian is rapidly becoming a recognised dialect, and scientific English and the spoken variety are not the same.
